# Histological and Molecular Biological Changes in Canine Skin Following Acute Radiation Therapy-Induced Skin Injury

**DOI:** 10.3390/ani14172505

**Published:** 2024-08-29

**Authors:** Sang-Yun Lee, Gunha Hwang, Moonyeong Choi, Chan-Hee Jo, Seong-Ju Oh, Yeung Bae Jin, Won-Jae Lee, Gyu-Jin Rho, Hee Chun Lee, Sung-Lim Lee, Tae Sung Hwang

**Affiliations:** 1College of Veterinary Medicine, Gyeongsang National University, Jinju 52828, Republic of Korea; sy_lee@gnu.ac.kr (S.-Y.L.); hgh3634@gmail.com (G.H.); ch_jo@gnu.ac.kr (C.-H.J.); osj414@gnu.ac.kr (S.-J.O.); ybjin@gnu.ac.kr (Y.B.J.); jinrho@gnu.ac.kr (G.-J.R.); lhc@gnu.ac.kr (H.C.L.); 2Yangsan S Animal Cancer Center, Yangsan 50638, Republic of Korea; vetmoon7@gmail.com; 3College of Veterinary Medicine, Kyungpook National University, Daegu 41566, Republic of Korea; iamcyshd@knu.ac.kr; 4Research Institute of Life Sciences, Gyeongsang National University, Jinju 52828, Republic of Korea; 5Institute of Animal Medicine, College of Veterinary Medicine, Gyeongsang National University, Jinju 52828, Republic of Korea

**Keywords:** canine, clinical change, quantitative real-time PCR, radiation therapy-induced injury

## Abstract

**Simple Summary:**

This study focused on understanding how radiation therapy, a common treatment for cancer, affects the skin of dogs. While radiation is effective at destroying cancer cells, it can also cause damage to healthy skin, leading to various side effects like redness, peeling, changes in skin color, and sores. Over nine weeks, we monitored these skin changes in dogs, observing that radiation led to increased inflammation and stress in the skin cells, as well as significant disruptions in how skin cells grow and heal. We also noticed changes in specific proteins and genes related to skin inflammation, healing, and cell death. These findings help us better understand how radiation therapy impacts the skin and provide valuable information for managing these side effects in dogs.

**Abstract:**

Radiation therapy is a crucial cancer treatment, but it can damage healthy tissues, leading to side effects like skin injuries and molecular alterations. This study aimed to elucidate histological and molecular changes in canine skin post-radiation therapy (post-RT) over nine weeks, focusing on inflammation, stem cell activity, angiogenesis, keratinocyte regeneration, and apoptosis. Four male beagles received a cumulative radiation dose of 48 Gy, followed by clinical observations, histological examinations, and an RT-qPCR analysis of skin biopsies. Histological changes correlated with clinical recovery from inflammation. A post-RT analysis revealed a notable decrease in the mRNA levels of Oct4, Sox2, and Nanog from weeks 1 to 9. VEGF 188 levels initially saw a slight increase at week 1, but they had significantly declined by week 9. Both mRNA and protein levels of COX–2 and Keratin 10 significantly decreased over the 9 weeks following RT, although COX–2 expression surged in the first 2 weeks, and Keratin 10 levels increased at weeks 4 to 5 compared to normal skin. Apoptosis peaked at 2 weeks and diminished, nearing normal by 9 weeks. These findings offer insights into the mechanisms of radiation-induced skin injury and provide guidance for managing side effects in canine radiation therapy.

## 1. Introduction

Radiation therapy stands as a prominent treatment modality for various types of cancer and is often utilized in conjunction with chemotherapy and surgery in the fields of both human and veterinary medicine [1,2]. While radiation therapy is highly effective in targeting and eradicating cancer cells, it is important to acknowledge that it can also impact healthy tissues, resulting in undesirable side effects and tissue damage [3]. Moreover, the interaction between the radiation beam and normal tissues can induce complex molecular biological changes, including alterations in oxidative stress, cell cycle regulation, and gene expression patterns, particularly in the skin [4,5,6].

While the primary objective of radiation therapy is the elimination of cancer cells, it frequently leads to unintended consequences, such as genomic instability in irradiated cells, which affects gene expression patterns [7,8]. Notably, the skin is one of the tissues most susceptible to radiation exposure. To investigate alterations in gene expression resulting from exposure to ionizing radiation, several research groups have examined human skin cells [9]. However, no prior studies have specifically explored changes in gene expression associated with radiation therapy for cancer treatment.

In the realm of veterinary medicine, radiation therapy is a valuable tool for managing cancer, yet it often brings about adverse effects, with skin damage being one of the most notable. Radiated healthy skin can exhibit a range of side effects, including erythema, desquamation, pigmentation changes, and ulceration [10]. These manifestations of radiation-induced skin injury are intricately linked to inflammatory responses and oxidative stress, which can lead to various forms of cell death, including mitotic death. Additionally, inflammation and oxidative stress can disrupt the cell cycle and cause DNA damage, potentially altering the cytokine profile [11]. To mitigate these effects, the use of anti-inflammatory agents and COX–2 inhibitors has been explored as a means of protecting normal cells from the side effects of radiation therapy [12]. In the application of RT, while it is important to predict the efficiency of the necrosis-like death of tumor cells [13] and mass size reduction, it is also crucial to understand the clinical and molecular biological mechanisms related to the occurrence, progression, and recovery process of RT-induced skin damage. Understanding these progression mechanisms of RT-induced skin damage will provide important clinical knowledge, such as establishing the application period of COX–2 inhibitors and skin recovery prognosis. 

To contribute to our understanding of the complexities surrounding post-irradiation reactions and the management of side effects associated with radiation therapy, it is imperative to comprehensively evaluate post-irradiated skin. In this study, we investigate the histological changes and alterations in mRNA levels of key factors relevant to cancer and stem cells (Oct4, Sox2, and Nanog) [14], angiogenesis and key aspects of tumorigenesis (VEGF 188) [15], inflammation (COX–2) [16,17], and keratinocyte differentiation (Keratin 10) [18,19,20] in canine skin tissue for 9 weeks following irradiation with a clinically applied therapeutic dose of radiation. Therefore, we will be able to understand the mechanism of radiation-induced skin injury by therapeutic dose by monitoring the specific process of inflammatory tissue damage and the recovery following keratinocyte differentiation. In addition, the investigation of changes in mRNA levels of core early stem cell transcription factors in skin tissue induced by irradiation will provide useful information on both RT for future cancer therapy, and the prediction of the regenerative ability and recovery of damaged skin tissue. Therefore, this study aims to provide valuable information on the molecular biological changes induced by radiation therapy in normal skin, and ultimately serves as a guideline for the effective management of the side effects of radiation therapy in dogs.

## 2. Materials and Methods

### 2.1. Ethics Approval and Animals

All procedures in this study received approval from the Research Ethics Committee of Gyeongsang National University Animal Center for Biomedical Experimentation (Approval Code: GNU-200916-D0061). For this study, we utilized four clinically healthy male beagles, aged 55 weeks and weighing approximately 8.7 kg.

### 2.2. Irradiation and Skin Biopsies

In preparation for anesthesia, all beagles underwent a 12 h fasting period. Anesthesia induction was accomplished through the intravenous administration of propofol (6 mg/kg; Myungmoon Pharm, Seoul, Korea), and to maintain general anesthesia, isoflurane (Hana Pharm, Seongnam-si, Gyeonggido, Korea) was administered with oxygen at a rate of 2 L/min, achieved through endotracheal intubation. Anesthesia-induced dogs were positioned in right lateral recumbency, and they were subjected to radiation targeting the left flank. Mega-voltage radiation therapy was administered using a linear accelerator (Elekta Synergy, Elekta AB, Stockholm, Sweden) with 6 MeV electrons. To enhance the distribution of radiation therapy, a 1 cm thick bolus was applied to cover the skin during irradiation. The treatment field covered an area measuring 14 cm × 14 cm, and each dog received a total of 12 fractions of 4 Gy, resulting in a cumulative dose of 48 Gy over the course of 12 days (d). Skin biopsies from the irradiated left trunk were performed under anesthesia at intervals of 1, 2, 3, 4, 5, 7, and 9 weeks after radiation exposure, following the same anesthesia protocol used during irradiation. For the control group, skin samples were obtained from the flank region that had not been subjected to radiation.

### 2.3. Extraction of RNA, cDNA Synthesis, and RT–qPCR

We analyzed the mRNA levels of POU domain class 5 transcription factor 1 (Oct4), sex-determining region Y box 2 (Sox2), and homeobox protein NANOG (Nanog) as core stem cell transcription factors, vascular endothelial growth factor 188 (VEGF 188) as the angiogenesis-related marker, prostaglandin-endoperoxide synthase 2 (COX–2) as the inflammation-related marker, and cytokeratin 10 (Keratin 10) as the keratinocyte differentiation-related marker in biopsied skin tissues. After all biopsied skin samples were collected, RNAs in canine skin tissue were extracted by the RNeasy Mini Kit (Qiagen, Redwood, CA, USA). To remove DNA contamination, an RNase-free DNase treatment step was performed. To use uncontaminated RNA, A260/280 ratios were evaluated by the OPTIZEN NanoQ Lite spectrophotometer before cDNA synthesis. Sample RNA of 1 μg was used with the Omniscript Reverse Transcription Kit (Qiagen) at 37 °C for 1 h to produce cDNA. RT–qPCR was performed using a Rotor-Gene (Qiagen) RT–qPCR machine with Rotor gene 2X SYBR Green mix (Qiagen), 2 μL cDNA per one reaction, and 0.7 μM forward and reverse primers. The RT–qPCR reaction was performed by the program under pre-denaturation, denaturation, and a combined annealing/extension step. 

Pre-denaturation was performed at 95 °C for 2 min; in total, 40 PCR cycles were performed at 95 °C for 10 s and 60 °C for 6 s. The melting curve shows that the temperature alters from 60 to 95 °C by 1 °C/s. According to the manufacturer’s instructions for the RT–qPCR program, the temperature was then decreased to 40 °C for 30 s. The CT values for each sample were analyzed by Rotor-Gene Q series software 2.1.0 (Qiagen, Germany). In addition, the relative levels of gene expression were calculated using the 2–ΔΔCT method. Table 1 lists the RT–qPCR primers.

### 2.4. Histological Assessment

The irradiated skin tissues were fixed in 4% formaldehyde overnight at 4 °C, and the fixed tissue samples were dehydrated and embedded in paraffin. To make slides, the embedded paraffin sample was cut into 5 μm sections, and the sections were subjected to deparaffinization. After deparaffinization, skin tissue sections were stained with hematoxylin and eosin (H&E) and analyzed for pathological morphology. Sections were observed under optical microscopy (Nikon Eclipse 80i, Tokyo, Japan), and images were scanned with a Photo Imaging System (Canon 600D, Tokyo, Japan).

### 2.5. Immunohistochemistry (IHC) Analysis

To analyze protein in the tissue section, the rehydrated sections were boiled in 0.1 M sodium citrate buffer (pH 6.0) for 10 min to retrieve the antigen. Then, 3% H_2_O_2_ was treated for 30 min at room temperature to remove the endogenous peroxidase in tissue, following protein blocking with 3% serum solution for 30 min at room temperature. After the blocking step, the slide was washed twice in PBS. The slides were incubated with primary antibodies, such as COX–2 (1:200, cat. no. Ma5-14568, Invitrogen, Carlsbad, CA, USA) and Cytokeratin 10 (1:100, cat. no. sc-70907, Santa Cruz, CA, USA) at 4 °C overnight. To identify the primary antibody, biotin-conjugated secondary IgG (1:200 Vector Laboratories, Newark, CA, USA) and an avidin–biotin–peroxidase complex (Elite ABC Kit; Vector Laboratories) were reacted. Under the manufacturer’s instructions, the HRP–DAB detection IHC kit was used for visualization. The sections were mounted, stained with hematoxylin as a counterstain, and then examined under a microscope. 

### 2.6. Terminal Deoxynucleotidyl Transferase dUTP Nick End Labeling (TUNEL) Assay

A TUNEL assay (Roche, Indianapolis, IN, USA) was performed to assess irradiated skin apoptosis. Following the manufacturer’s instructions, the sections were stained with the TUNEL reaction mixture for 1 h in a humidified chamber in the dark at 37 °C. The slides were then stained with hematoxylin as a counterstain, and they were examined by microscope. At least 4 random stained slides per sample were estimated by ImageJ with the IHC profiler plugin. In the plugin option, IHC macro was performed to quantify the immunoreaction as a log score of high positive, positive, low positive, negative, final core, and also by histogram. 

### 2.7. Statistical Analysis

Data were analyzed using PASW Statistics 18 (SPSS Inc., Chicago, IL, USA), and were presented as the mean ± SEM. A one-way ANOVA with Newman–Keuls post hoc test was used to determine the statistical differences, which were regarded as significant when *p* < 0.05.

## 3. Results

### 3.1. Observable Change of Radiation-Induced Skin Injury

Radiation was administered to the left trunk skin of the beagles, totaling 48 Gy delivered 12 times at 4 Gy/d over a span of 12 d, with continuous monitoring of their health conditions. Skin changes after radiation were observed for 9 weeks after the end of the radiation therapy, and significant and progressive radiation-induced skin reactions were observed (Figure 1). Following immediate irradiation (Figure 1A), a mild erythema was observed in the radiation-exposed skin, while no significant changes in the systemic physical condition were noted. However, post-irradiation symptoms, such as moist desquamation, wounds, and scaling of the skin, were observed on the skin from one week after irradiation (Figure 1B). After 2–3 weeks post-irradiation, a significant alopecia/epilation change was observed on the irradiation area of the skin (Figure 1C,D). Then, generalized erythema was alleviated, but hyperpigmentation of the irradiated area of skin was developed at 4–5 weeks (Figure 1E,F). Radiation-induced hyperpigmentation deposition in the irradiated skin progressed gradually, with increased hyperpigmentation observed at 7–9 weeks (Figure 1G,H).

### 3.2. Histological Changes of Skin Tissues for 9 Weeks Post-RT

The histological changes of the irradiated skin tissues were compared with normal non-irradiated skin tissues for 9 weeks after irradiation (Figure 2). We observed the histological changes progressing from inflammation to histological recovery in irradiated skin. First, at 1 week post-irradiation, hemorrhage was observed by the destruction of muscle structure and vessels in the skin, compared to the histologically normal skin; also, interruptions of the epidermis and nuclear fragments were observed (Figure 2B). Then, progressive morphological changes, such as epidermal hyperplasia, interface dermatitis, and hyperkeratosis were observed at 2–4 weeks post-irradiation (Figure 2C–E). After 5 weeks of radiation therapy, the skin still displayed mild fibrosis and hyperkeratosis (Figure 2F), and the changes in pigmentation were detected in 7–9-week post-irradiated skin (Figure 2G,H).

### 3.3. Expression of Cancer Stem Cells and Angiogenesis-Related Factors Post-RT

We analyzed changes in the mRNA expression levels of core factors of cancer and stem cell-related Oct4, Sox2, and Nanog and angiogenesis-related VEGF 188, after exposure to RT by an RT–qPCR analysis (Figure 3). Compared to normal skin tissue, Oct4, Sox2, and Nanog were significantly (*p* < 0.05, *p* < 0.01, and *p* < 0.0001) decreased in irradiated skin at 1 week after RT, and the reduced levels were maintained until 9 weeks after RT. VEGF 188 was increased at 1 week post-RT, but was found to be significantly (*p* < 0.01 and *p* < 0.0001) downregulated compared to normal skin from 2 to 9 weeks post-RT.

### 3.4. Expression of Inflammation and Keratinocyte Differentiation Markers Post-RT

To estimate changes in the skin healing process after RT exposure, we analyzed the expression of inflammation-related COX–2 and keratinocyte differentiation-related Keratin 10 by RT–qPCR and immunohistochemical staining. mRNA levels of COX–2 were significantly (*p* < 0.0001) increased at 1 to 2 weeks in post-irradiation skin (Figure 4A), and protein levels of COX–2 also increased at the same period post-RT, which was confirmed by the increased positive signal by immunohistochemical staining (Figure 4C). As a keratinocyte differentiation ability in the skin, the mRNA level of Keratin 10 was significantly (*p* < 0.01) decreased at 1 to 2 weeks post-RT, but increased after 3 weeks, and dramatically increased at 4 weeks post-RT. Then, the mRNA levels of Keratin 10 were stepwise decreased after 5 weeks of RT (Figure 4B). Keratin 10 positive signals were mainly found from 3 to 9 weeks, being excluded at 1 to 2 weeks in post-RT skin (Figure 4D); therefore, the protein level of Keratin 10 was shown to be consistent with the tendency of mRNA levels.

### 3.5. Radiation-Induced Apoptosis in Irradiated Skin Tissues

Radiation-induced apoptosis was assessed in skin tissue post-RT by the immunohistochemical staining of the TUNEL assay (Figure 5). For the quantification of immunohistochemical staining, scores of high positivity, positivity, low positivity, and negativity were assigned as 4, 3, 2, and 1, respectively (Figure 5A). Apoptosis scores increased throughout the entire period, except for 7 to 9 weeks post-RT, and increased significantly (*p* < 0.05 and *p* < 0.0001) at 2 to 5 weeks after RT, compared with normal skin. These apoptosis scores decreased to a similar level compared to normal skin at 9 weeks after RT.

## 4. Discussion

In veterinary oncology, radiotherapy stands as a pivotal treatment modality for cancer [21], offering meaningful benefits in managing cancers throughout whole life stages, including senior and geriatric canine patients when invasive treatments become challenging. However, while radiotherapy has a killing effect on target cancer cells, it also has a destructive effect on normal tissue cells in the irradiation field, especially in producing moderate-to-severe skin reactions. Radiation therapy can cause side effects, such as radiation dermatitis [22], and radiation-induced skin injuries reduce the quality of life for both canine patients and their owners, while also exerting economic pressures for the treatment of additional skin damage. Changes in oxidative stress and cell cycle arrest have been reported to be associated with radiation-induced skin injury and various aspects of molecular biology [23], and this regulatory mechanism can be related to the recovery property of irradiated normal skin around cancer tissue. To the best of our knowledge, there has been no study of the mechanisms of histological and molecular biological changes accompanied by clinical changes in normal canine skin induced by RT exposure. Therefore, due to the adverse effects of radiotherapy at the molecular level, an assessment of irradiated skin tissue changes is important to understand the complexity of the post-irradiation response and the treatment of adverse effects in veterinary clinical care. 

In this study, we monitored various dermatological changes for 9 weeks after irradiating the skin with a therapeutic dose of radiation commonly applied clinically for cancer treatment in veterinary medicine. After nine weeks of observation following irradiation, clinical dermatological assessments revealed progressive radiation-induced skin reactions. These included symptoms such as erythema, moist desquamation, scarring, and alopecia, alongside radiation dermatitis accompanied by pigmentation changes. These skin reactions appear to have been caused by the epidermis and radiation-sensitive tissue, while acute radiation dermatitis, which usually occurs within 90 d, is reported to be the most common reaction to radiation therapy [22]. In particular, these radiation-induced skin injuries undergo clinical and histological changes, such as erythema, desquamation, pigmentation, and ulceration due to radiation treatment conditions that require repeated irradiation [10]. The histological changes of irradiated skin were similarly correlated with clinical observations that were progressive from inflammation at 1 week to histological recovery from 2 to 9 weeks in irradiated skin (Figure 1 and Figure 2). In particular, after 1 week, irradiated skin showed hemorrhage, nuclei fragments, epidermal hyperplasia, fibrosis, and hyperkeratosis. This histological change was consistent with a previous study that stated that radiation injuries, such as epidermal hyperplasia, interface dermatitis, loss of hair follicles, hyperkeratosis, and dermal fibrosis, were characteristic of morphological change [24]. Therefore, irradiating normal canine skin tissues with therapeutic doses may lead to radiation dermatitis, inducing clinical and histological changes in the epidermis, and causing radiation-induced skin damage. Our study demonstrated the correlation between macroscopic and microscopic changes in the skin following radiation therapy. This suggests that, in clinical practice, macroscopic changes observed after radiation therapy can indicate corresponding microscopic alterations. Consequently, when macroscopic lesions are detected in the skin after radiation therapy, clinicians can anticipate the underlying histological changes and implement appropriate clinical interventions and management strategies.

We analyzed the molecular biology of factors involved in the skin healing process, including inflammation, wound healing, and keratinocyte regeneration in the irradiated skin. Such an analysis facilitates predictions of both clinical and histological changes in radiation-induced skin reactions, and it is able to suggest an understanding of potential therapeutic approaches. In this study, we investigated the molecular biological changes that inevitably accompany clinical changes in irradiated skin suffering from radiation-induced dermatitis. Therefore, the mRNA levels of cancer- and stem cell-related transcription factors (Oct4, Sox2 and Nanog), angiogenesis (VEGF 188), inflammation (COX–2), and keratinocyte differentiation (Keratin 10)-related factors were analyzed in biopsied skin at 9 weeks after irradiation, and HPRT1 was adopted as the most suitable reference gene for the normalization of mRNA levels in radiation-exposed canine dermal tissues [25]. Transcription factors Oct4, Sox2, and Nanog are the most important core factors to maintain the pluripotency of stem cells [14]. The expression patterns of these transcription factors could predict the pluripotency potential of adult stem cells and resident mesenchymal stem cells within skin tissue exposed to radiation, and they may enable foresight of the regenerative potential of irradiated skin. In particular, Oct4, Sox2, and Nanog are overexpressed in malignancies, and affect tumor transformation and metastasis [26]. Since cancer stem cells have the potential to form a tumor, the inactivation of all cancer stem cells following radiation therapy is necessary to achieve long-term local and distant tumor control. However, it was shown that cancer stem cells survive many commonly used cancer therapies with resistance [27]. VEGF is the most powerful inducer of angiogenesis and is necessary for tumor invasion and metastasis; the predictive significance of VEGF in canine malignant tumors has been reported [15]. In the present study, the expression of Oct4, Sox2, and Nanog was significantly decreased in all skin tissues for 9 weeks post-RT. This is consistent with previous studies that showed the decrease in cancer stem cell factors, such as Oct4, Sox2, and Nanog due to cancer therapy, including chemoradiotherapy or radiation therapy [28,29]. In addition, this study observed both a reduction in cancer stem cell factors and fibrosis in post-radiation skin. It was considered that radiation therapy in normal skin induced fibrosis and decreased stem cell factors. Previously, after radiation exposure, stem cells differentiated into myofibroblast cells, and decreased Oct4, Sox2, and Nanog [30]. At 1 week post-radiation therapy of skin, VEGF 188 was a little increased, compared with normal skin. This result was in accordance with a previous study revealing that the expression of VEGF following 2 Gy irradiation was slightly increased [31]. In addition, numerous studies have demonstrated that radiation elevated VEGF expression in a variety of normal tissues and cancers in vivo and in vitro at different time points, ranging from 1 h to 20 weeks, following different doses of irradiation of 0.1–75 Gy [32,33]. VEGF 188 was significantly decreased at 2, 3, 4, 5, 7, and 9 weeks post-radiation therapy, compared with normal tissue. The result of VEGF 188 downregulation was considered to be skin wounding caused by radiation-induced injury. In a previous study, the expression of VEGF decreased gradually over time after wounding [34]. An initial increase in VEGF 188 appears to facilitate the recovery from the early inflammatory response and skin barrier disruption. Then, VEGF decreases over time as part of the wound healing process. Together, these results show that radiation therapy affected cancer stem cell factors and skin degeneration such as fibrosis, which was due to the loss of progenitors and wounding in post-irradiated skin tissue. 

VEGF is significantly associated with a high expression of COX–2 in canine malignant mammary tumors [35]. COX–2 promotes angiogenesis and cell proliferation, encourages metastatic spread, and contributes to tumor-associated inflammation and mammary tumorigenesis. In particular, COX–2 participates in several cancer hallmarks, while the COX–2/prostaglandin E2 (PGE2) pathway plays an important role in carcinogenesis [36], both in humans [37,38,39] and dogs [16,17]. In response to damage, inflammation, and cell activation by a variety of agonists, such as cytokines and growth factors, COX–2 and Keratin 10 are changed in various tissues, including the epidermis. UV irradiation, which causes temporary or persistent hyperplasia, such as epidermal malignancies, can cause COX–2 expression in the epidermis [40,41,42]. When the wound healing process and tissue regenerate, Keratin 10, which is a crucial structure protein, regulates dermal cell proliferation, differentiation, and migration [18,19,20]. In our study, the expression level of COX–2 and Keratin 10 in mRNA and protein was investigated for 9 weeks post-RT. For 2 weeks post-RT, the expression of COX–2 in mRNA and protein levels was significantly increased, compared with normal skin. This upregulation of the expression of COX–2 was consistent with a previous study on acute ultraviolet irradiation in the hairless SKH–1 mouse [43]. Four weeks post-RT, COX–2 expression was returned to normal. Previous studies reported that COX–2 inhibitors have radioprotective and radiosensitizing effects [44,45]. Based on our findings, it would be reasonable to use COX–2 inhibitors to manage radiation-induced dermatitis before 4 weeks post-RT. After 1 and 2-week radiation therapy, the expression of Keratin 10 in mRNA and protein levels was significantly decreased, compared with normal skin. The decreased expression of Keratin 10 is suggested to be caused by radiation-induced skin injury. Previously, expressions of Keratin 10–Keratin 1 in suprabasal keratinocytes downregulate during skin injury [46]. After 4 weeks post-irradiation, the expression of Keratin 10 was upregulated. In accordance with our histological assessment, it was shown that hyperkeratosis was caused by the upregulation of Keratin 10. It was demonstrated in a previous study that the hyperkeratotic epidermis was caused by the increased expression of Keratin 10 [47]. In conclusion, through the expression of COX–2 and Keratin 10, our study showed changes in epidermal markers at each week post-RT.

The apoptosis induction of skin is one of the features for radiation-induced dermatitis [48], and this radiation-induced apoptosis is directly and indirectly related to DNA damage. When the radiation damage is irreparable, the apoptosis pathways are activated [49]. In our study, post-irradiated skin was analyzed for apoptosis by a TUNEL assay and scored using an IHC profiler. Post-RT, the apoptosis score was increased during the whole period, excluding 9 weeks. The apoptosis score peaked at 2 weeks, then gradually decreased. These results indicate that apoptosis was reduced due to the recovery of acute radiation dermatitis. This is consistent with a previous study that showed that as the post-irradiated skin damage was repaired, the percentage of apoptotic cells decreased [48]. Our study results indicate that apoptosis persists in normal canine skin up to 7 weeks post-RT. Therefore, it is recommended to follow up on the radiation side effects on the skin until 9 weeks post-RT, when the apoptosis score decreased in our study. There are also reports of using stem cell therapy to reduce the side effects of radiation therapy, such as apoptosis and dermatitis [50]. Our results are expected to provide guidelines on initiation and duration to apply treatment for radiation side effects.

However, it is important to note that our study observed changes only up to 9 weeks post-irradiation. Long-term effects beyond this period, including dermal fibrosis, atrophy, dyspigmentation, telangiectasia, and the long-term changes in biomarkers, were not assessed. Further research is necessary to evaluate these long-term histological and molecular changes.

In this study, we examined the irradiated skin tissue changes at a molecular level. However, we only estimated gene and protein expression change on the treatment of radiation side effects. We need to complete a more comprehensive analysis of the biological pathways and mechanisms affected by the observed biomarker changes. In addition, we have only examined male beagles. It is necessary to investigate the irradiated skin tissue changes in other species and sexes. Thus, further studies are required to provide a more comprehensive understating of irradiated skin change across different species and sexes.

## 5. Conclusions

This study investigated the histological and molecular biological changes in canine skin following radiation therapy (RT) over a nine-week period. By examining the inflammatory responses, stem cell activity, angiogenesis, keratinocyte regeneration, and apoptosis, significant insights were gained into the mechanisms of radiation-induced skin injury and the subsequent recovery process. We found that cancer stem cell factors and VEGF were suppressed, and their expression levels were not able to recover until 9 weeks after irradiation. Further, COX–2, Keratin 10, and apoptosis were changed for the progression of dermatitis, epidermal hyperplasia, and irradiated skin damage. We suggest the application of COX–2 inhibitor for the first 1–2 weeks after irradiation. Skin recovery from radiation-induced dermatitis is expected to be reached 9 weeks after irradiation by the recovery of similar levels of keratinocyte differentiation and apoptosis to those of normal skin. However, long-term side effects after the 9-week period were not assessed, necessitating further long-term follow-up research. Our study provides a theoretical foundation for therapeutic approaches to manage radiation-induced skin reactions in canine cancer treatment.

## Figures and Tables

**Figure 1 animals-14-02505-f001:**
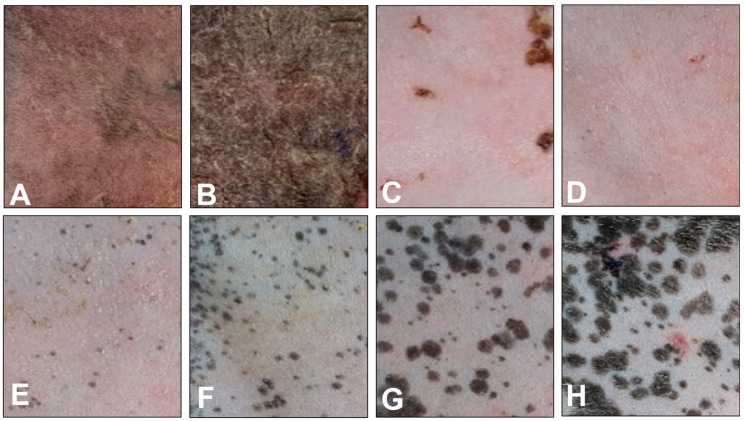
RT exposure of therapeutic doses to normal skin and clinical change observations of the irradiated area for 9 weeks. Observation of clinical change in the irradiated skin area (**A**) immediately post-RT; (**B**) 1 week; (**C**) 2 weeks; (**D**) 3 weeks; (**E**) 4 weeks; (**F**) 5 weeks; (**G**) 7 weeks; and (**H**) 9 weeks after irradiation.

**Figure 2 animals-14-02505-f002:**
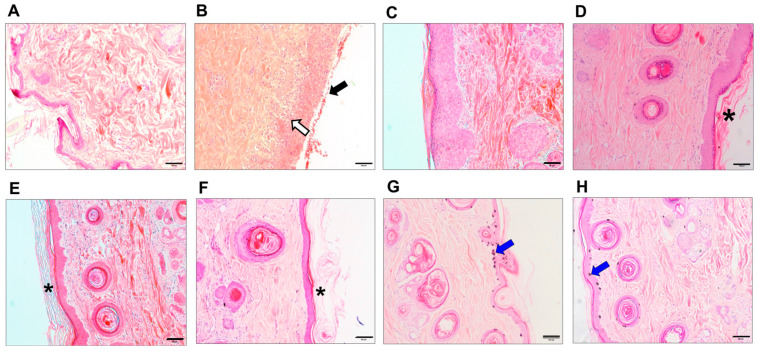
A histopathological examination of biopsy skin tissues using H&E staining for 9 weeks post-RT. After radiation exposure, several morphological changes with specific structures were observed in the irradiated skin tissues. (**A**) normal skin; (**B**) 1 week; (**C**) 2 weeks; (**D**) 3 weeks; (**E**) 4 weeks; (**F**) 5 weeks; (**G**) 7 weeks; and (**H**) 9 weeks post-RT. The tissues show hemorrhage (black arrow), nuclei fragment (white arrow), pigmentation (blue arrow), and hyperkeratosis (asterisk). The images are representative of each group (scale bar = 100 μm).

**Figure 3 animals-14-02505-f003:**
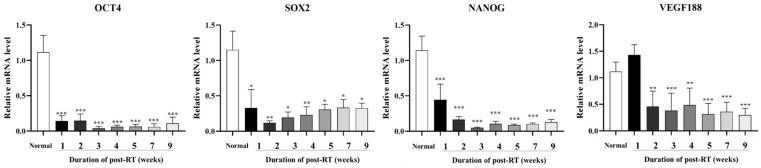
Expression levels of stem cell-related transcription factors (Oct4, Sox2, and Nanog) and VEGF 188 for 9 weeks post-RT. Normal skin was used as a control, and mRNA levels were analyzed from 1 to 9 weeks after irradiation. The data are presented as the mean ± SEM. * *p* < 0.05, ** *p* < 0.01, *** *p* < 0.0001.

**Figure 4 animals-14-02505-f004:**
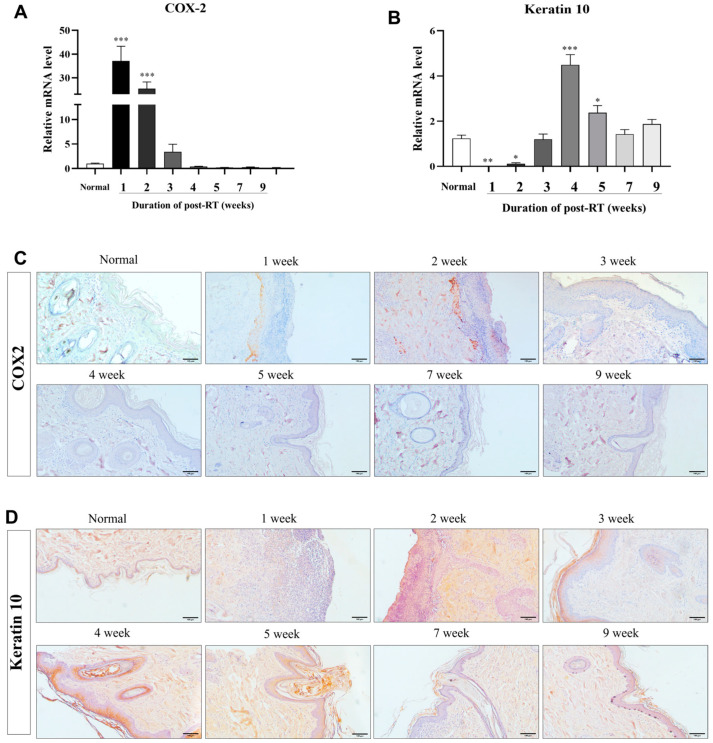
Expression levels of factors relevant to inflammation (COX–2) and keratinocyte differentiation markers (Keratin 10) for 9 weeks post-RT. RT–qPCR and IHC stain were used to analyze mRNA and protein levels at 1, 2, 3, 4, 5, 7, and 9 weeks post-irradiation. (**A**,**B**) Changes in mRNA levels of COX–2 and Keratin 10 in skin tissue 9 weeks after irradiation through an RT–qPCR analysis. (**C**,**D**) An observation of histopathological changes of COX–2 and Keratin 10 in skin tissues for 9 weeks after irradiation through IHC staining. Normal skin is unirradiated skin tissue, presented as the control group. The images are representative of each group (scale bar = 100 μm). The data were presented as the mean ± SEM. * *p* < 0.05, ** *p* < 0.01, *** *p* < 0.0001.

**Figure 5 animals-14-02505-f005:**
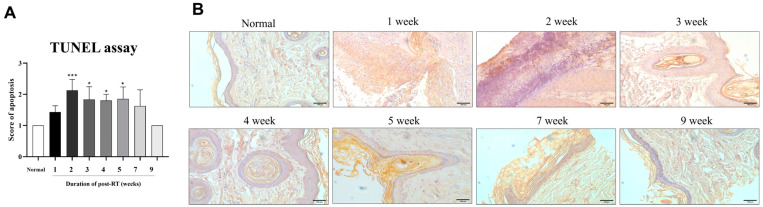
An assessment of apoptosis in skin tissues for 9 weeks post-RT by TUNEL staining. (**A**) Scoring of Apoptosis (TUNNEL assay) in tissue estimated by ImageJ with an IHC profiler plugin. (**B**) An observation of the apoptosis signal in irradiated skin tissues through TUNEL staining. The images are representative of each group (scale bar = 100 μm). The data are presented as the mean ± SEM. * *p* < 0.05, *** *p* < 0.0001.

**Table 1 animals-14-02505-t001:** Primer pairs used for RT–qPCR.

Gene Name (Symbol)	Primer Sequences	Amplicon Size (bp)	Accession Number
POU domain class 5 transcription factor 1 (Oct4)	F: AACGATCAAGCAGTGACTATTCG R: AGTAGAGCGTAGTGAAGTGAGG	147	NM_001003142.2
Sex determining region Y box 2 (Sox2)	F: AGTCTCCAAGCGACGAAAAA R: CCACGTTTGCAACTGTCCTA	189	DR105272
Homeobox protein NANOG (Nanog)	F: GACCGTCTCTCCTCTTCCTTCC R: CGTCCTCATCTTCTGTTTCTTGC	157	XM_014108418.1
Vascular endothelial growth factor 188 (VEGF 188)	F: CGAGTACATCTTCAAGCCATCC R: GTGATGTTGAACTCCTCAGTGG	102	AF133250.1
Prostaglandin–endoperoxide synthase–2 (COX–2)	F: TGTTCACCTGACTACTGGAAGC R: GACAGCCCTTCACGTTATTGC	109	NM_001003354.1
Cytokeratin 10 (Keratin 10)	F: CTCGTGACTACAGCAAATACTACC R: TGGCATTGTCGATCTGAAGC	105	NM_001013425.1
Hypoxanthine phosphoribosyltransferase 1 (HPRT1)	F: GACTGAAGAGCTACTGTAATGACC R: TCTTTGGATTATGCTCCTTGACC	168	XM538830.1

## Data Availability

The data presented in this study are available within the article.

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
