# Peer review of "Histological and Molecular Biological Changes in Canine Skin Following Acute Radiation Therapy-Induced Skin Injury"

_animals, 2024, doi:10.3390/ani14172505_

Round 1

Reviewer 1 Report

Comments and Suggestions for Authors

This is an interesting study reviewing the gross, histological, and molecular features of radiation-induced skin damage. This is a common side-effect of radiation therapy and this study serves to detail these findings. This could be utilized in the future to guide clinical support when this issue arises as a consequence of treatment.

The immunohistochemistry methodology section does not list catalog numbers for the antibodies utilized. Please include these, as this enables replication in the future. Companies will often carry multiple antibodies with the same target, hence, catalog numbers are vital for identification of the utilized antibody.

Lines 207, 221, and 309: Please clarify what RBC means. If reb blood cells, the wording on a few sentences can be improved. E.g. in line 209 "RBC resulting in hemorrhage was observed..." -> "hemorrhage was observed..."

Line 287: "other side effects of various molecular biology,..." -> please edit this sentence for clarity.

Line 358: Are you able to expand on the reasoning for an initial increase in VEGF 188 followed by a decrease? The discussion mostly mentions these results, but fails to address what is the impact of this finding.

396: the percentage of apoptosis cells -> the percentage of apoptotic cells

401: Please edit the opening statement of the conclusion for clarity.

407: While some of the examined markers trended towards normal levels by week 9, both the discussion and conclusion fail to highlight the duration as a limitation of this study. Animals were followed until 9 weeks, hence stating that "recovery is achieved in 9 weeks" is slightly misleading. Longer term effects (dermal fibrosis and contraction) could easily take longer than 9 weeks to fully develop.

Comments on the Quality of English Language

Overall, the introduction, materials, discussion, and conclusion require few changes. Editing of English language for clarity would be beneficial, especially in the discussion as there are a few conflicting sentences. I highlighted a few instances above of sentences that could be clarified.

Author Response

Comments and Suggestions for Authors

This is an interesting study reviewing the gross, histological, and molecular features of radiation-induced skin damage. This is a common side-effect of radiation therapy and this study serves to detail these findings. This could be utilized in the future to guide clinical support when this issue arises as a consequence of treatment.

â–º We sincerely thank you for your efforts in reviewing our manuscript. We have included our point-by-point responses to your comments below.

- The immunohistochemistry methodology section does not list catalog numbers for the antibodies utilized. Please include these, as this enables replication in the future. Companies will often carry multiple antibodies with the same target, hence, catalog numbers are vital for identification of the utilized antibody.

Response: We appreciate the reviewers’ valuable opinions We have revised the manuscript to include information on catalog numbers for the antibodies utilized. (Materials and Methods section, page 4, line 160 to 161)

: COX−2 (1:200, cat. no. Ma5-14568, Invitrogen, CA, USA) and Cytokeratin 10 (1:100, cat. no. sc-70907, Santa Cruz, CA, USA))

- Lines 207, 221, and 309: Please clarify what RBC means. If reb blood cells, the wording on a few sentences can be improved. E.g. in line 209 "RBC resulting in hemorrhage was observed..." -> "hemorrhage was observed..."

Response: We appreciate the reviewers’ valuable opinions. RBC refers to red blood cells. We have revised the sentence to improve clarity. The corrections have been made accordingly (Result section, page 6 and 9, line 207, 221 and 309) 

 - Line 287: "other side effects of various molecular biology,..." -> please edit this sentence for clarity.

Response: We appreciate the reviewers’ valuable comment. To clarity the sentence, we have revised it. The corrections have been made accordingly (Discussion section, page 8, line 286 to 288) 

: Changes in oxidative stress and cell cycle arrest have been reported to be associated with radiation-induced skin injury and various aspects of molecular biology

- Line 358: Are you able to expand on the reasoning for an initial increase in VEGF 188 followed by a decrease? The discussion mostly mentions these results, but fails to address what is the impact of this finding.

Response: We appreciate the reviewers’ valuable question. In our study, canine skin exposed to radiation showed signs of inflammation and histological evidence of skin barrier disruption. To facilitate the recovery of irradiated skin, there appears to be a temporary increase in VEGF 188, which may be related to previous findings (Reference 31) that indicated an increase in VEGF due to mast cell degradation. As VEGF levels decrease, both visible and histological changes suggest a wound healing process, consistent with results from a previous study (Reference 34). We have addressed the reviewer’s opinion in the discussion section. (Discuss section, page 10, line 361 to 363)

: An initial increase in VEGF 188 appears to facilitate the recovery from the early inflammatory response and skin barrier disruption. Then, VEGF decreases over time as part of the wound healing process.

- 396: the percentage of apoptosis cells -> the percentage of apoptotic cells

Response: We appreciate the reviewers’ valuable opinions. The corrections have been made accordingly (Discuss section, page 10, line 404) 

- 401: Please edit the opening statement of the conclusion for clarity.

Response: We appreciate the reviewers’ valuable opinions. The corrections have been made accordingly (conclusion section, page 11, lines 424-428) 

: This study investigated the histological and molecular biological changes in canine skin following radiation therapy (RT) over a nine-week period. By examining the inflammatory responses, stem cell activity, angiogenesis, keratinocyte regeneration, and apoptosis, significant insights were gained into the mechanisms of radiation-induced skin injury and the subsequent recovery process.

- 407: While some of the examined markers trended towards normal levels by week 9, both the discussion and conclusion fail to highlight the duration as a limitation of this study. Animals were followed until 9 weeks, hence stating that "recovery is achieved in 9 weeks" is slightly misleading. Longer term effects (dermal fibrosis and contraction) could easily take longer than 9 weeks to fully develop.

Response: We appreciate the reviewers’ valuable opinions. The necessary corrections have been made accordingly. We have added a statement in the Discussion section to highlight that our study reports on changes observed up to 9 weeks, and we have noted the need for further research on long-term changes in the Limitations and Conclusion sections. (Discussion section, page 11, lines 411-415; Conclusion section, page 12, lines 432-433)

: However, it is important to note that our study observed changes only up to 9 weeks post-irradiation. Long-term effects beyond this period, including dermal fibrosis, atrophy, dyspigmentation, telangiectasia, and the long-term changes in biomarkers, were not assessed. Further research is necessary to evaluate these long-term histological and molecular changes.

: However, long-term side effects after the 9-week period were not assessed, necessitating further long-term follow-up research.

Reviewer 2 Report

Comments and Suggestions for Authors

Review of the Paper: "Histological and Molecular Biological Changes in Canine Following Acute Radiation Therapy-induced Skin Injury"

Strengths:

The approaches provide a comprehensive analysis of the skin's reaction to radiation, including clinical observations, histological evaluations, and molecular studies (RT-qPCR and immunohistochemistry).

The full scope of the biological alterations is offered by the employment of several biomarkers (e.g., COX-2, Keratin 10, Oct4, Sox2, Nanog, VEGF 188) in the investigation of inflammation, stem cell activity, and keratinocyte differentiation.

The method used in this work to measure the histological and molecular alterations during the nine weeks following radiation therapy contributes to a deeper comprehension of the course and recuperation from radiation-induced damage.

Replicability and ethical compliance are provided by the thorough explanation of the methodology and ethical issues, such as radiation dosages and anesthesia protocols.

Recommendations:

A more comprehensive analysis of the biological pathways and mechanisms affected by the observed biomarker changes is essential. This includes detailing the specific roles of VEGF in angiogenesis and COX-2 in inflammation within the context of radiation-induced damage.

The exclusive use of male beagles in the study limits the generalizability of the results. Future research must address potential sex-specific differences in the response to radiation therapy by including both male and female subjects.

The statistical methods used for analyzing RT-qPCR and immunohistochemistry data, particularly in selecting and normalizing reference genes, need to be clearly outlined to ensure reproducibility.

The study's focus on a nine-week post-radiotherapy period is insufficient; further research should explore the long-term stability and relevance of these biomarkers in chronic phases of radiation-induced damage.

The brief mention of potential therapeutic implications in the paper is inadequate. A detailed discussion is required on how these findings can be translated into clinical practice, including potential therapeutic interventions or preventative measures for managing radiation-induced skin damage in both human and veterinary medicine.

Overall Assessment:

Understanding the histological and molecular alterations that follow radiation therapy in canine models is greatly advanced by this work. It offers important baseline data for veterinary oncology research in the future as well as possible clinical applications. Expanding the demographic scope and delving further into biological causes may augment the study's efficacy.

Author Response

Reviewer 2

Comments and Suggestions for Authors

Review of the Paper: "Histological and Molecular Biological Changes in Canine Following Acute Radiation Therapy-induced Skin Injury"

Strengths:

The approaches provide a comprehensive analysis of the skin's reaction to radiation, including clinical observations, histological evaluations, and molecular studies (RT-qPCR and immunohistochemistry).

The full scope of the biological alterations is offered by the employment of several biomarkers (e.g., COX-2, Keratin 10, Oct4, Sox2, Nanog, VEGF 188) in the investigation of inflammation, stem cell activity, and keratinocyte differentiation.

The method used in this work to measure the histological and molecular alterations during the nine weeks following radiation therapy contributes to a deeper comprehension of the course and recuperation from radiation-induced damage.

Replicability and ethical compliance are provided by the thorough explanation of the methodology and ethical issues, such as radiation dosages and anesthesia protocols.

â–º We sincerely thank you for your efforts in reviewing our manuscript. We have included our point-by-point responses to your comments below.

Recommendations:

- A more comprehensive analysis of the biological pathways and mechanisms affected by the observed biomarker changes is essential. This includes detailing the specific roles of VEGF in angiogenesis and COX-2 in inflammation within the context of radiation-induced damage.

Response: We appreciate the reviewers’ valuable opinions. We recognize the need for a comprehensive analysis of the biological pathways and mechanisms, and as such, we are considering this for further study. Therefore, we have mentioned the limitations of this study. (Discussion sections, page 10 to 11, line 416 to 419)

: In this study, we examined the irradiated skin tissue changes at molecular level. However, we only estimated gene and protein expression change on treatment of radiation side effects. We need to more comprehensive analysis of the biological path-ways and mechanisms affected by the observed biomarker changes.

- The exclusive use of male beagles in the study limits the generalizability of the results. Future research must address potential sex-specific differences in the response to radiation therapy by including both male and female subjects.

Response: We appreciate the reviewers’ valuable comment. We have also been considering sex-specific differences. Also, it is important to investigate changes not only across different sexes but also in other species. We are therefore preparing further studies on this matter and have mentioned this point in the discussion section. (Discussion sections, page 10 to 11, line 419 to 422)

: In addition, we have only examined male beagles. It is necessary to investigate for the irradiated skin tissue changes in other species and sex. Thus, further studies are required to provide a more comprehensive understating of irradiated skin change across different species and sex.

- The statistical methods used for analyzing RT-qPCR and immunohistochemistry data, particularly in selecting and normalizing reference genes, need to be clearly outlined to ensure reproducibility.

Response: We appreciate the reviewers’ valuable opinions. In our previous study, we confirmed that HPRT1 is the most stable reference gene in irradiated skin tissue using geNorm, Normfinder and bestKeeper. In addition, We confirmed that using unstable reference genes for normalization resulted in different statistical significance (Reference 25). To ensure reproducibility, we selected the stable reference gene and performed normalization using the same method in this study. (Discussion section, page 9, line 324 to 330)

- The study's focus on a nine-week post-radiotherapy period is insufficient; further research should explore the long-term stability and relevance of these biomarkers in chronic phases of radiation-induced damage.

Response: We appreciate the reviewers’ valuable opinions. The necessary corrections have been made accordingly. We have added a statement in the Discussion section to highlight that our study reports on changes observed up to 9 weeks, and we have noted the need for further research on long-term changes in the Limitations and Conclusion sections. (Discussion section, page 11, lines 411-415; Conclusion section, page 12, lines 432-433)

: However, it is important to note that our study observed changes only up to 9 weeks post-irradiation. Long-term effects beyond this period, including dermal fibrosis, atrophy, dyspigmentation, telangiectasia, and the long-term changes in biomarkers, were not assessed. Further research is necessary to evaluate these long-term histological and molecular changes.

: However, long-term side effects after the 9-week period were not assessed, necessitating further long-term follow-up research.

- The brief mention of potential therapeutic implications in the paper is inadequate. A detailed discussion is required on how these findings can be translated into clinical practice, including potential therapeutic interventions or preventative measures for managing radiation-induced skin damage in both human and veterinary medicine.

Response: We appreciate the reviewers’ valuable opinions. The necessary corrections have been made accordingly. We have additionally mentioned the clinical significance, treatment implications, and research relevance at the end of each paragraph in the Discussion section. (Discussion section, page 9, lines 315-321; page 10, lines 383-386; page 11, lines 404-407)

: Our study demonstrated the correlation between macroscopic and microscopic changes in the skin following radiation therapy. This suggests that, in clinical practice, macroscopic changes observed after radiation therapy can indicate corresponding microscopic alterations. Consequently, when macroscopic lesions are detected in the skin after radiation therapy, clinicians can anticipate the underlying histological changes and implement appropriate clinical interventions and management strategies.

: . Previous studies reported that COX-2 inhibitors have radioprotective and radiosensitizing effects. [44, 45]. Based on our findings, it would be reasonable to use COX-2 inhibitors to manage radiation-induced dermatitis before 4 weeks post-RT

: Our study results indicate that apoptosis persists in normal canine skin up to 7 weeks post-RT. Therefore, it is recommended to follow up on the radiation side effects on the skin until 9 weeks post-RT, when the apoptosis score decreased in our study.

Round 2

Reviewer 2 Report

Comments and Suggestions for Authors

By providing a detailed analysis of the biological pathways and mechanisms impacted by the observed biomarker changes, acknowledging the limitations of using only male beagles, and clarifying the statistical methods for analyzing RT-qPCR and immunohistochemistry data, the authors have successfully addressed all previous concerns. They have also highlighted the necessity for long-term research on biomarker stability beyond the nine-week post-radiotherapy period and expanded on the therapeutic implications of their findings. The inclusion of these revisions has significantly improved the manuscript. The paper is a valuable contribution to the field, and I recommend its publication due to these enhancements.

Author Response

Comments and Suggestions for Authors

Review of the Paper: "Histological and Molecular Biological Changes in Canine Following Acute Radiation Therapy-induced Skin Injury"

â–º We sincerely thank you for your efforts in reviewing our manuscript. We have included our point-by-point responses to your comments below.

Line 2 presumably 'canine skin' not just 'canine'?

Response: We appreciate the reviewers’ valuable opinions. The corrections have been made accordingly (line 2)

: Histological and Molecular Biological Changes in Canine Skin Following Acute Radiation Therapy-induced Skin Injury

Simple summary. The Simple Summary is not in layperson's language. This is an important part of the paper that simplifies the main findings for a lay readership. Imagine you are trying to describe your work to a random person on the street, this is your challenge, and it needs to be much better than this. Please see recent issues of Animals for examples of Simple Summaries.

Response: We appreciate the reviewers’ valuable comment. We have revised the text using layperson's language. (simple summary, line 23-30)

: This study focused on understanding how radiation therapy, a common treatment for cancer, affects the skin of dogs. While radiation is effective at destroying cancer cells, it can also cause damage to healthy skin, leading to various side effects like redness, peeling, changes in skin color, and sores. Over nine weeks, we monitored these skin changes in dogs, observing that radiation led to increased inflammation and stress in the skin cells, as well as significant disruptions in how skin cells grow and heal. We also noticed changes in specific proteins and genes related to skin inflammation, healing, and cell death. These findings help us better understand how radiation therapy impacts the skin and provide valuable information for managing these side effects in dogs.
